# Comparison of the Effects of Metformin and Thiazolidinediones on Bone Metabolism: A Systematic Review and Meta-Analysis

**DOI:** 10.3390/medicina59050904

**Published:** 2023-05-08

**Authors:** Ru-Dong Chen, Cong-Wen Yang, Qing-Run Zhu, Yu Li, Hai-Feng Hu, Da-Chuan Wang, Shi-Jie Han

**Affiliations:** 1Shandong Provincial Hospital Affiliated to Shandong First Medical University, Jinan 250021, China; chenrudong19971027@163.com (R.-D.C.); zhuqingrun@163.com (Q.-R.Z.);; 2Department of Neurosurgery, Weifang Medical University, Weifang 261042, China; ycw995089435@163.com

**Keywords:** metformin, thiazolidinediones, diabetes, osteoporosis, bone density, bone turnover markers

## Abstract

Objectives: Studies have shown that people with diabetes have a high risk of osteoporosis and fractures. The effect of diabetic medications on bone disease cannot be ignored. This meta-analysis aimed to compare the effects of two types of glucose-lowering drugs, metformin and thiazolidinediones (TZD), on bone mineral density and bone metabolism in patients with diabetes mellitus. Methods: This systematic review and meta-analysis were prospectively registered on PROSPERO, and the registration number is CRD42022320884. Embase, PubMed, and Cochrane Library databases were searched to identify clinical trials comparing the effects of metformin and thiazolidinediones on bone metabolism in patients with diabetes. The literature was screened by inclusion and exclusion criteria. Two assessors independently assessed the quality of the identified studies and extracted relevant data. Results: Seven studies involving 1656 patients were finally included. Our results showed that the metformin group had a 2.77% (SMD = 2.77, 95%CI [2.11, 3.43]; *p* < 0.00001) higher bone mineral density (BMD) than the thiazolidinedione group until 52 weeks; however, between 52 and 76 weeks, the metformin group had a 0.83% (SMD = −0.83, 95%CI: [−3.56, −0.45]; *p* = 0.01) lower BMD. The C-terminal telopeptide of type I collagen (CTX) and procollagen type I N-terminal propeptide (PINP) were decreased by 18.46% (MD = −18.46, 95%CI: [−27.98, −8.94], *p* = 0.0001) and 9.94% (MD = −9.94, 95%CI: [−16.92, −2.96], *p* = 0.005) in the metformin group compared with the TZD group.

## 1. Introduction

Diabetes is a very common chronic metabolic disorder marked by hyperglycemia. Currently, diabetes affects more than 387 million adults worldwide [1]. This figure is expected to reach 592 million by 2035 [1]. The prevalence of diabetes is increasing at an alarming rate. Epidemiological data indicate that the incidence of osteoporosis and fractures triggered by increased bone fragility in patients with diabetes is already significantly higher than in the healthy population [2,3,4]. Type 1 (T1DM) and type 2 diabetes (T2DM) are both associated with bone disease, although by different mechanisms [5]. In T1DM, reduced bone mass and bone strength lead to a five-fold increase in the risk of fractures. In T2DM, the risk of fractures is increased despite normal bone mass [6]. The negative effects of hyperglycemia that lead to advanced glycosylation end products (AGEs) on bone tissue may also contribute to the development of osteoporosis [3,7]. 

Not only does blood sugar hurt bones, glucose-lowering medications exert direct or indirect effects on bones as well, which cannot be ignored [4,8,9,10]. There has been much evidence that thiazolidinediones (TZDs) reduce bone mineral density and thus increase fracture risk [11,12]. This may be related to the function of TZDs, including reducing osteoclast-specific transcription factor activity and osteoblast-specific signaling pathway activity [13,14]. In contrast, metformin has a potentially protective effect on bone tissue. Metformin positively affects osteoblast differentiation by activating the AMPK pathway and reversing the deleterious effects of advanced glycosylation end products (AGEs) [9,14]. However, in two vitro studies, metformin did not affect the osteogenic differentiation of bone marrow mesenchymal stem cells and the stromal mineralization of primary osteoblasts [15]. High concentrations of metformin may even significantly inhibit osteoblast differentiation [16]. In some clinical trials, metformin has been shown to reduce the risk of fractures in patients with diabetes. Conversely, other clinical trials have shown no significant correlation between metformin and hip fracture in patients with T2DM, or that metformin exerts harmful effects on the bone [17]. Some studies have also attempted to explain this paradox by suggesting that metformin is usually used in patients with less severe T2DM or of shorter duration. At the early stages of T2DM, blood glucose damage to the bone is less severe and patients are at less risk of fractures [18]. Moreover, most clinical trials have not considered the effect of the drug’s glucose-lowering efficacy on bone tissue.

The most common method for identifying osteoporosis is the quantitative measurement of bone marrow density (BMD) by the T-score assay. However, BMD is not particularly sensitive as a monitoring tool for treatment response, as changes in density may be slow or small [19]. Bone turnover markers (BTMs) have become promising tools in the management of osteoporosis because they provide dynamic information on bone status and can be used as an alternative to BMD testing; BTMs may be especially useful in monitoring osteoporosis treatment responses [20]. In this study, the primary outcome measures are BMD and BTMs. BTMs are consistent standard markers for assessing bone resorption (C-terminal telopeptide of type I collagen [CTX]) and formation (procollagen type I N-terminal propeptide [PINP]) [21,22]. BTMs can reflect bone remodeling in diabetic bone disease, assess fracture risk, and monitor osteoporosis. Moreover, BTMs are used to assess osteoporosis treatment responses. In contrast, bone-specific alkaline phosphatase (BAP) and urine n-telopeptide of type I collagen (u-NTX) are negatively correlated with BMD [23].

In summary, we searched electronic databases to identify clinical trials comparing the effects of metformin and TZDs on bone metabolism in patients with diabetes. In our meta-analysis, we compared BMD and BTMs in the metformin and TZD groups to determine whether bone loss due to uncoordinated bone formation and bone resorption may be caused by the administration of glucose-lowering drugs. Our goal was to elucidate the effects of two glucose-lowering drugs on bone metabolism and to contribute new approaches to bone conservation in diabetic patients.

## 2. Materials and Methods

The protocol for this review is registered on PROSPERO (International prospective register of systematic reviews) and the registration number is CRD42022320884 (Appendix A). All procedures relevant to a systematic review were performed in line with the Preferred Reporting Items for Systematic Reviews and Meta-Analyses (PRISMA) guidelines [24]. The PRISMA 2020 checklist [25] is shown in Appendix A.

### 2.1. Search Strategy

The PubMed, Embase, and Cochrane Library databases were searched to identify studies written in the English language in the last fifteen years. The following MeSH terms were used for the search terms: (metformin) and (thiazolidinediones or pioglitazone or rosiglitazone) and (bone metabolism or bone biomarkers or osteoporosis or bone density). Randomized controlled trials (RCTs) published up to 20 April 2023 were included based on our inclusion and exclusion criteria. The search strategies are presented in Figure 1.

### 2.2. Inclusion and Exclusion Criteria

#### 2.2.1. Participants

The inclusion criteria were as follows: (a) patients, aged ≥18 years, with diabetes; (b) patients with either T1DM or T2DM.

The exclusion criteria were as follows: (a) patients with clinically significant combined liver disease, renal impairment, history of lactic acidosis, unstable or severe angina pectoris, heart failure, uncontrolled hypertension, etc.; (b) patients intolerant to metformin or thiazolidinediones; (c) patients with a long history of anti-osteoporosis medication use.

#### 2.2.2. Interventions and Exposures

1.Clinical intervention trials comparing thiazolidinediones and metformin.2.Combined insulin for blood glucose control.

#### 2.2.3. Study Types

The inclusion criteria were as follows: (a) published high-quality randomized controlled trials (RCTs); (b) studies with a follow-up of at least 80%, which include BMD or BTM as primary outcomes; (c) studies with complete treatment outcomes. 

The exclusion criteria were as follows: (a) non-clinically controlled trials and low-quality trials (studies without randomization, blinding, allocation hiding, and processing of incomplete data); (b) reviews, animal studies, studies that were not relevant to the question, data that were not extractable, or the study results are not yet published; (c) interventions and outcome indicators that are inconsistent with our study.

### 2.3. Outcomes

The primary outcome measures of this study are BMD and the percentage change in BTMs (procollagen type I N-terminal propeptide [PINP] and C-terminal telopeptide of type I collagen [CTX]).

The secondary outcomes are bone-specific alkaline phosphatase [BAP] and urine n-telopeptide of type I collagen (u-NTX).

### 2.4. Data Extraction

Two researchers extracted studies from the database searches. For the included studies, relevant data including the first authors, publication date, location, sample size, gender, age, patient characteristics, interventions, and outcomes were collected [26]. For incomplete data, the authors were contacted, and any missing data were obtained. Controversial data were discussed until a consensus was reached, or a third investigator was consulted to resolve any disagreements.

### 2.5. Risk of Bias and Quality Assessment

The systematic review and meta-analysis were conducted according to the methods published in the Cochrane Handbook [27]. The methodological quality of the included RCTs was assessed according to the Cochrane Handbook for Systematic Reviews of Interventions [28] (Figure 2 and Figure 3).

### 2.6. Statistical Analysis and Assessment of Publication Bias

Review Manager Software (Version 5.3, Nordic Cochrane Center, London, UK) was used to perform the meta-analysis. Heterogeneity was assessed using the I^2^ statistic (I^2^ < 50% and *p* > 0.1 was considered low heterogeneity). Due to different patient characteristics, treatment options, and other factors, potential heterogeneity was unavoidable. According to the Cochrane Handbook, it is always preferable to explore possible causes of heterogeneity. In this study, there were too few articles to enable an evaluation of heterogeneity. Random-effects meta-analysis can be used to integrate heterogeneity among studies. Compared with a fixed-effects meta-analysis, a random-effects meta-analysis gives more weight to smaller studies. This approach allows us to address heterogeneity that is not easily explained by other factors. Therefore, we applied the random effects model to small studies with potentially high heterogeneity. The weighted mean difference (WMD) or standardized mean difference (SMD) was used to assess the continuous outcomes with 95% confidence intervals [CI] [36]. Any *p* values < 0.05 were considered statistically significant. 

For data presenting 95% confidence intervals rather than standard deviations, the standard deviations (SD) were calculated according to the formula (x − 1.96 × SD/√n, x + 1.96 × SD/√n).

## 3. Results

### 3.1. Search Results

We searched the Embase, PubMed, and Cochrane Library databases. In the initial search, we identified 127 articles. After removing duplicates, 62 articles remained. Based on the titles and abstracts, we excluded reviews (*n* = 7) (including systematic reviews and meta-analyses) and articles that did not correspond to the content of the study (*n* = 16); 39 articles were found to be relevant. We analyzed the full text of the 39 articles to exclude studies that were not clinically controlled trials (*n* = 17) and interventions that did not test metformin and TZDs (*n* = 15). Finally, 7 RCTs, with 1656 patients, were identified as having met the inclusion criteria [29,30,31,32,33,34,35]. The search strategies are presented in Figure 1. Table 1 summarizes the characteristics of the seven included studies.

All articles reported the generation of randomized sequences. Of the seven studies included in the meta-analysis, five reported blinding of outcome assessors and clinical staff. Three studies concealed the allocation scheme. All included studies reported outcomes for >95% of participants (Figure 2 and Figure 3). In Bilezikian’s study, a 52-week double-blind phase (rosiglitazone or metformin) was followed by a 24-week open-label phase, during which all patients received metformin.

### 3.2. Primary Outcomes

#### 3.2.1. Changes in BMD

Two studies (*n* = 291) were included in the meta-analysis [29,32]. The results of our meta-analysis showed a 2.77% increase in BMD at 52 weeks in the metformin group compared with the TZD group, and the result was statistically significant (SMD = 2.77, [95%CI 2.11, 3.43]; *p* < 0.00001; Figure 4). However, during weeks 52–76, the metformin group showed a 2% decrease in BMD compared with the TZD group (SMD = −0.83, [95%CI −3.56, −0.45] *p* = 0.01; Figure 4); again, the result was significant. In Bilezikian’s study, at weeks 0–52, the BMD of the femoral neck and lumbar spine increased by 0.22% and 0.04%, respectively. Conversely, the BMD of the total hip decreased by 0.72% in the metformin treatment group. In the treatment group, the BMD of the femoral neck, total hip, and lumbar spine decreased (1.47%, 1.62%, and 1.41%, respectively). At weeks 52–76, the BMD of the femoral neck and total hip decreased by 0.02% and 0.13%, respectively, in the metformin group, while the lumbar spine BMD increased by 1.03% compared with the baseline. After switching to metformin at 24 weeks for patients who previously took RSG, the total hip and lumbar spine BMD increased by 0.4% and 0.26%, respectively. The femoral neck BMD decreased to 0.07% compared with the baseline but showed an upward trend compared to week 52. Furthermore, the data from Miller (2016) are hip cortical vBMD (volumetric bone mineral density, measured by peripheral quantitative computed tomography). Bilezikian (2013) reported the areal BMD measured by dual-energy X-ray absorptiometry (DXA).

#### 3.2.2. Changes in BTMs

C-terminal telopeptide of type I collagen *(CTX)* Four studies, involving 1320 patients, reported percentage changes in the CTX [30,31,32,33]. The meta-analysis showed a significant difference in CTX (MD = −18.46, 95%CI: [−27.98, −8.94], *p* = 0.0001; Figure 5a) between metformin and TZDs. Compared with the TZD group, CTX in the metformin group decreased by 18.46%. In four included studies, CTX was reduced in the metformin group compared with the TZD group. CTX decreased in all metformin groups compared to pre-administration, and in the TZDs group, a 1% decrease was observed only in Zinman’s study of men.

Procollagen type 1 N-propeptide (P1NP) Three studies, involving 1262 patients, reported percentage changes in the BTM, and P1NP [30,31,33]. The meta-analysis showed a significant difference in P1NP (MD = −9.94, 95%CI: [−16.92, −2.96], *p* = 0.005; Figure 5b). Compared with the TZD group, P1NP in the metformin group decreased by 9.94%. Three studies reported changes in P1NP, except for Lierop’s study in which P1NP was increased in patients treated with TZDs. The other studies showed a decreasing trend in P1NP regardless of whether metformin or TZDs were administered.

#### 3.2.3. Secondary Outcomes

Bone alkaline phosphatase (BAP) Two studies with 1158 patients reported the BAP [30,34]. Our results showed that BAP did not differ significantly between patients on metformin and those on TZD (MD = −2.69, 95%CI: [−7.11, 1.74], *p* = 0.23; Figure 6A).

Urine n-telopeptide of type I collagen (u-NTX). Two studies of 103 patients were included in the meta-analysis. The change in u-NTX did not differ significantly in patients on metformin vs. those on TZD (MD = −3.15, 95%CI: [−14.76, 8.46], *p* = 0.59; Figure 6B) [34,35]. 

## 4. Discussion

Based on the included studies, the control of glucose metabolism among patients on metformin vs. TZD was similar. However, differences in bone mineral density and bone turnover markers were evident. Our results indicate that the two types of drugs have different effects on bone metabolism that are independent of their glucose-lowering effects. Our meta-analysis found that BMD was higher in the metformin group than in the TZD group by week 52, while the opposite effect was noted from weeks 52 to 76. Moreover, the BTMs, PINP, and CTX were reduced in the metformin group compared with the TZD group. Furthermore, the BAP and u-NTX levels did not differ significantly between the two treatment groups.

There is evidence that TZD treatment can lead to bone loss and increase the risk of osteoporosis [11]. In contrast, the use of metformin may reduce the risk of osteoporosis and fractures [37]. Our findings in this review conflict with the current evidence. During the first 52 weeks of dosing, the metformin group had a 2.77% higher BMD than the TZD group; however, between 52 and 76 weeks, the metformin group had a 0.83% lower BMD than the TZD group. Notably, this trend of increasing and then decreasing BMD with metformin was compared with the TZD group, and BMD was elevated in the metformin group relative to the TZD group before week 52. The higher BMD in the metformin group compared with that of the TZD group in the first 52 weeks of treatment may be due to the more pronounced inhibitory effect of TZD on osteogenesis. Furthermore, in both trials included in this meta-analysis, compared to baseline, the lumbar spine and femoral neck BMD was increased, while the hip BMD decreased in the metformin group [29,32]. Based on the results of this current study, it is not clear whether metformin itself has a promotive effect on osteogenesis. At weeks 52–76, BMD decreased more in the metformin group than in the TZD group, and all included studies showed a decrease. In the long term, metformin did not have a positive effect on preventing bone loss, which contradicts the findings from previous studies which reported an osteoprotective effect in those treated with metformin. Given the limited studies included, this finding should be considered with caution. Although most of the evidence indicates a significant negative association between metformin use and fracture risk [3,33,38], most of the available evidence was drawn from studies that only analyzed fractures at 6 months, while the effect of blood glucose reduction on bone tissues in those treated with metformin was not considered [11].

Several recent large cohort studies and retrospective analyses support our conclusions. The results from a cohort study that included 64,878 patients with T2DM found that there was no significant association between metformin use and hip fractures [39]. Another retrospective analysis reported that the reduced fracture risk associated with metformin may be due to reduced fracture risks in the patients indicated for the drug. Specifically, metformin is commonly prescribed for the early stages of T2DM. The association between metformin use and a reduced fracture risk due to time-related biases cannot be completely excluded, which could lead to spurious treatment benefits [18]. Therefore, more rigorously designed studies with longer observation periods are needed to evaluate the effect of metformin on BMD in the future.

CTX and P1NP are recommended by the International Osteoporosis Foundation/International Federation of Clinical Chemistry (IOF/IFCC) as the preferred markers for monitoring osteoporosis [40]. They are consistent standard markers for evaluating bone resorption (CTX) and formation (P1NP) about fracture risks and osteoporosis monitoring. Most of the relevant research showed that metformin promotes osteogenesis, while TZDs induce bone loss [3,10,41]. The results of our meta-analysis showed that CTX and P1NP decreased by 18.46% and 9.94%, respectively, in the metformin group compared with the TZD group. 

CTX is a collagen fragment produced during bone resorption [42], which can be quantified in serum and/or urine by specific immunoassays and is used as a clinical marker for osteoclastic activities [43]. In this review, CTX, as a marker of bone resorption, was decreased by 18.46% in the metformin group compared with the TZD group. Moreover, CTX was decreased in the metformin group in all four of the included studies. In contrast, CTX in the TZD group was decreased only in male patients in Zinman’s study, while other studies demonstrated an increase. These findings indicate that metformin significantly inhibited osteoclastic activities compared with TZDs. It has been shown that metformin decreases the receptor activator of nuclear factor kappa-B ligand expression and increases osteoprotegerin expression in osteoclasts, thereby inhibiting osteoclast differentiation, leading to a decreased number of osteoclasts, and preventing bone loss [44]. In hematopoietic cells, metformin inhibits the development of osteoclasts [45]. The effect of TZDs on osteoclast activities is currently unclear.

P1NP, as a marker of osteogenesis, represents the formation of new bone, and the level of P1NP reflects the exact osteoblastic activities [46]. The higher the level of P1NP in the blood, the more active the osteoblasts. Our findings showed that of the studies included, only Lierop’s study showed an increase in P1NP levels in the TZD group, while the other studies all demonstrated a decrease. In contrast, P1NP decreased in all metformin groups. The results of our meta-analysis showed a 9.94% decrease in P1NP levels in the metformin group compared with the TZD group. The result indicates that both metformin and TZDs inhibit osteogenesis. Importantly, metformin has a stronger effect on osteogenesis inhibition than TZDs. Current evidence suggests that TZDs increase the risk of fractures [47,48], but the mechanism by which TZDs increase the risk of fractures is unclear. It may be related to the activation of peroxisome proliferator-activated receptor-γ (PPAR-γ), which differentiates mesenchymal stem cells into adipocytes and reduces osteoblast differentiation at the cellular level [15]. In patients with diabetes, metformin can reduce the deleterious effects of hyperglycemia and AGEs on osteoblasts [49,50]. Moreover, metformin can induce osteoblast proliferation and differentiation by activating AMPK, ERK-1/2, and e/iNOS, thus promoting osteogenesis in vitro [15]. However, it has also been shown that metformin has no osteogenic effect in rodents [51].

The pathogenesis leading to osteoporosis is highly complex. A prominent characteristic of osteoporosis is the increase in bone turnover. The decrease in both CTX and PINP after metformin treatment suggests that there is an overall decrease in bone turnover but that metformin is not anabolic to bone tissues. This also supports the findings from other studies in which metformin was found to reduce osteoporosis. This effect may be related to a decrease in bone turnover.

PINP and BAP respond to osteoblast activities and are closely associated with bone formation in patients with osteoporosis [52]. Bone resorption markers are usually degradation products of bone collagen (NTX) molecules; CTX, which is released into the circulation and excreted in the urine, reflects the resorptive activity of osteoclasts [52]. The results of our meta-analysis showed (Figure 6) that the differences in BAP and u-NTX between the metformin group and the TZD group were not significant. However, BAP decreased in the metformin group in all three studies, indicating that metformin did not promote osteogenesis. Similarly, u-NTX decreased in the metformin group in both studies, suggesting that metformin also did not promote bone resorption. This was consistent with the previous conclusion that both PINP and CTX decreased in the metformin group.

This study has some limitations as follows. (1) This meta-analysis included only seven studies. Our analysis would have been more credible if more RCTs had been included. (2) We cannot exclude the possibility that the changes in BTM levels may be influenced by other factors, such as nutrition, exercise, and liver function. (3) We included only two studies involving BMD. Due to the small sample size, the BMD results need to be interpreted with caution. (4) The exact relationship between BTMs and osteogenic and osteolytic activities requires further investigation. In the future, clinical studies that explore the physiological mechanisms of BTMs are necessary. (5) In the heterogeneity test, BMD, CTX, and P1NP showed high heterogeneity (I2 > 50%). Even though we used a random effects model to reduce the effect of heterogeneity, the effect of higher heterogeneity on the results of the meta-analysis could not be eliminated due to the small number of included studies. Therefore, we should interpret these observations with caution and look forward to more high-quality clinical studies in the future to remedy this deficiency.

## 5. Conclusions

Our results showed that metformin does not promote bone anabolism in patients with diabetes. TZDs inhibited osteogenesis and promoted osteolysis. The short- and long-term effects of metformin and TZDs on BMD warrant further investigations. Evidence from available studies confirms that metformin can inhibit osteoclasts, and the positive effect of metformin on osteoporosis may be achieved through the inhibition of osteoclast activities. Many studies have shown that TZD inhibits osteogenesis, leading to an increased risk of osteoporosis in patients with diabetes. However, based on our analysis, metformin inhibits osteoblasts more significantly than TZD. Hence, the effects of these drugs on osteoporosis in patients with diabetes need to be further examined.

## Figures and Tables

**Figure 1 medicina-59-00904-f001:**
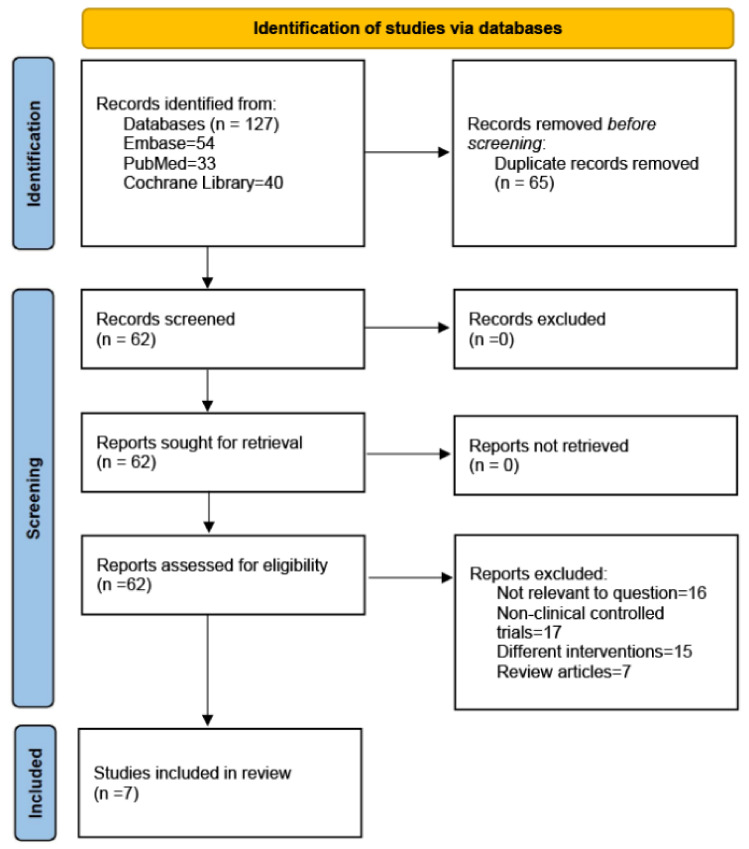
PRISMA 2020 flow diagram.

**Figure 2 medicina-59-00904-f002:**
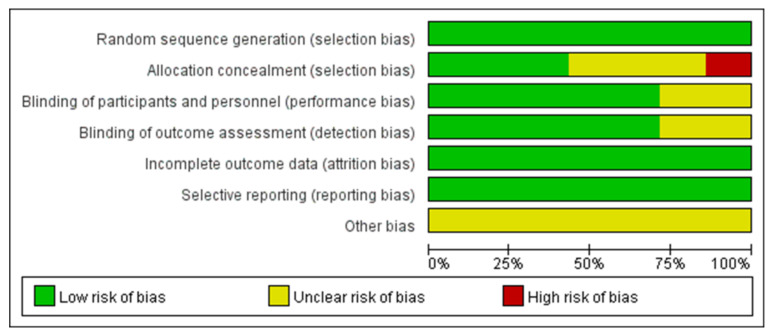
A graph depicting the risk of bias.

**Figure 3 medicina-59-00904-f003:**
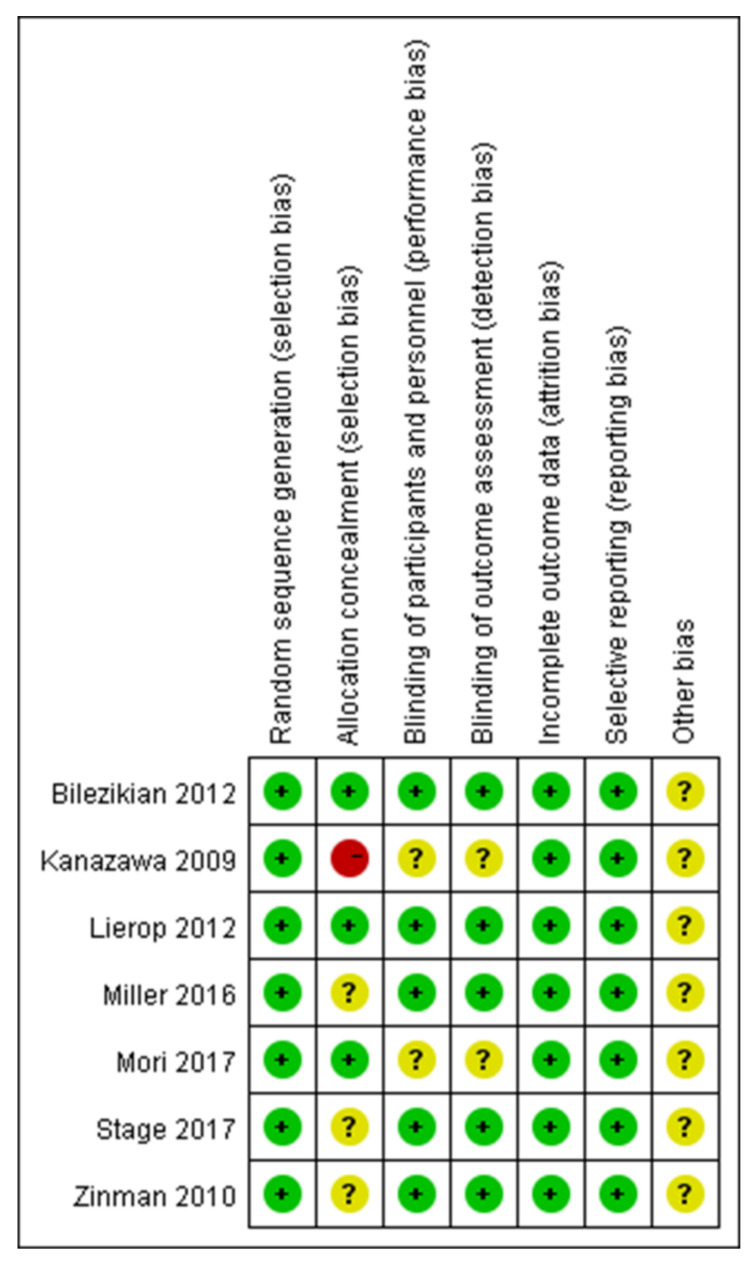
A risk of bias summary [29,30,31,32,33,34,35].

**Figure 4 medicina-59-00904-f004:**
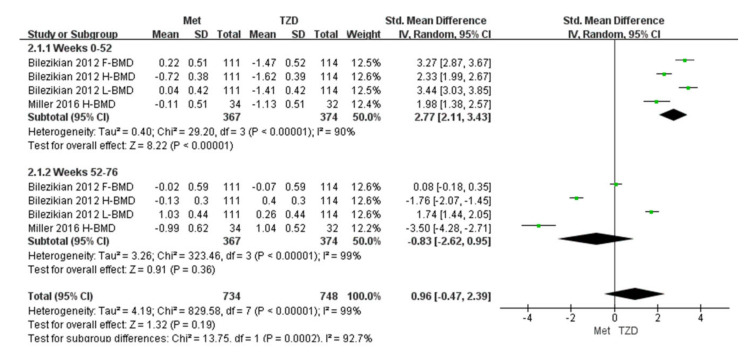
Forest plot for comparing the BMD of metformin and TZD. The green squares represent the weight of each study [29,32].

**Figure 5 medicina-59-00904-f005:**
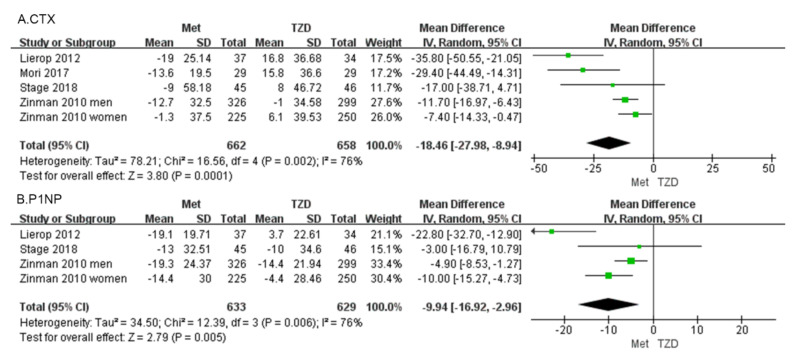
Forest plot comparing metformin and TZD and BTMs, CTX (**A**) and P1NP (**B**). The green squares represent the weight of each study [30,31,32,33].

**Figure 6 medicina-59-00904-f006:**
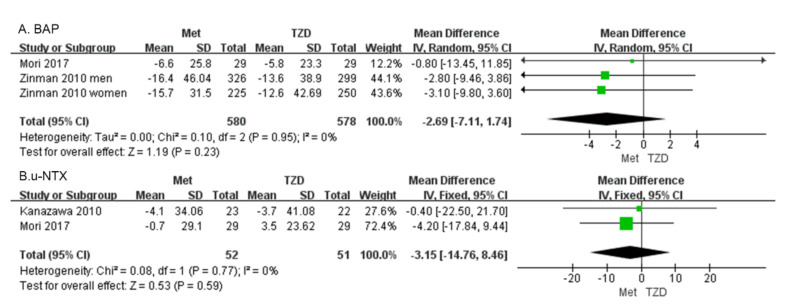
Forest plot comparing BAP (**A**) and u-NTX (**B**) in patients on metformin vs. those on TZD. The green squares represent the weight of each study [30,34].

**Table 1 medicina-59-00904-t001:** Study and patient characteristics.

Study	Year	Country	Sample Met/TZD	Age Met/TZD	Gender Women%	BMI (kg/m^2^) Met/TZD	Intervention	Daily Dose Met/TZD	Patients	Observed Duration	Study Design
Zinman	2010 [30]	America and Europe	551/549	56.6 ± 9.4/ 56.9 ± 10.0	40.8%/ 45.5%	32.8 ± 6.1/ 32.5 ± 6.5	Met/ Rosiglitazone	2 g/8 mg	T2DM	12 Months	RCT
Lierop	2012 [31]	The Netherlands	37/34	56.5 ± 5.4/ 56.4 ± 5.9	0%	29.3 ± 3.8/ 28.2 ± 3.0	Met/ Pioglitazone	1 g/15 mg	T2DM	24 Weeks	RCT
Miller	2016 [32]	USA	34/32	55.0 ± 16.4/ 56.5 ± 5.6	100%	25.4 ± 4.0/ 28.7 ± 3.4	Met/Rosiglitazone	2 g/8 mg	T2DM	76 Weeks	RCT
Stage	2017 [33]	Denmark	45/46	57	38%	34.6	Met/Rosiglitazone	1 g/8 mg	T2DM	24 Weeks	RCT
Mori	2017 [34]	Japan	29/29	65.1 ± 7.7/ 64.1 ± 8.5	55.17%/ 62.07%	24.3 ± 3.8/25.6 ± 4.0	Met/Pioglitazone	750 mg/ 15 or 30 mg	T1DM	3 Months	RCT
Kanazawa	2009 [35]	Japan	23/22	66 ± 10/ 67 ± 10	43.5%/ 36.3%	24.9 ± 3.7/22.0 ± 2.3	Met/Pioglitazone	500–750 mg/15–30 mg	T2DM	3 Months	RCT
Bilezikian	2012 [29]	USA	111/114	64.0 ± 6.46/ 63.6 ± 6.61	100%	31.5 ± 5.79/ 31.2 ± 5.86	Met/Rosiglitazone	2g/8 mg	T2DM	76 Weeks	RCT

Met: metformin, TZD: thiazolidinedione, T2DM: type 2 diabetes mellitus, T1DM: type 1 diabetes mellitus, NM: no mention, RCT: randomized controlled trials.

## Data Availability

The data supporting the conclusions of this article are available online.

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
