# Peer review of "Comparison of the Effects of Metformin and Thiazolidinediones on Bone Metabolism: A Systematic Review and Meta-Analysis"

_medicina, 2023, doi:10.3390/medicina59050904_

Round 1
Reviewer 1 Report (Previous Reviewer 2)
Abstract: Registration of the meta-analysis should be placed in the methods section. For the percentage mentioned in the results, they should be accompanied by 95% CI.
Introduction: line 48: … reduce bone mineral density and increase fracture risk.
Methods section 2.5: PRISMA checklist is the standard reporting guideline for meta-analysis. It is not used for appraising the studies included.
Results: The results for BMD and bone remodelling markers showed high level of heterogeneity as indicated by I2>50%. However, the authors did not attempt to resolve these issues. This makes the analysis less convincing.
Line 160-161: Bilezikian (2013) is the true bone mineral density measured by DXA… what does this sentence mean?
Discussion: line 204: … our study… please avoid this expression as the data from other studies.
Overall, the grammar of the manuscript needs to be revised vigorously. The use of abbreviations needs to be standardised (eg: Met, TZD, BMD, BAP, u-NTX). Please do not capitalise generic drug names.
Author Response
Comments:
- Abstract: Registration of the meta-analysis should be placed in the methods section. For the percentage mentioned in the results, they should be accompanied by 95% CI.
Answer:We are very sorry to have omitted this. We put the registration of the meta-analysis in the methods section. For the percentages in the results add the 95% CI.
- Introduction: line 48: … reduce bone mineral density and increase fracture risk.
Answer:The corresponding modifications were made. (Page 3, Line 53)
- Methods section 2.5: PRISMA checklist is the standard reporting guideline for meta-analysis. It is not used for appraising the studies included.
Answer:Thank you very much for your suggestion. We have modified this sentence as “All procedures relevant to a systematic review were performed in line with the Preferred Reporting Items for Systematic Reviews and Meta-Analyses (PRISMA) guideline”. (Page 6, Line 129-132)
- Results: The results for BMD and bone remodelling markers showed high level of heterogeneity as indicated by I2>50%. However, the authors did not attempt to resolve these issues. This makes the analysis less convincing.
Answer:According to the Cochrane Handbook, it is always preferable to explore possible causes of heterogeneity. In this study, there were too few articles to enable an evaluation of heterogeneity. Random-effects meta-analysis can be used to integrate heterogeneity among studies. Compared to a fixed-effects meta-analysis, a random-effects meta-analysis gives more weight to smaller studies. This approach allows us to address heterogeneity that is not easily explained by other factors. Therefore, we applied the random effects model to small studies with potentially high heterogeneity.(Page7, Line143-149)
- Line 160-161: Bilezikian (2013) is the true bone mineral density measured by DXA… what does this sentence mean?
Answer: We are very sorry to have made this mistake. It is not “true bone mineral density” but “areal bone mineral density”.
- Discussion: line 204: … our study… please avoid this expression as the data from other studies.
Answer:Thank you for your helpful comments. This expression has been corrected to “our meta-analysis showed that” or “based on our analysis”.
- Overall, the grammar of the manuscript needs to be revised vigorously. The use of abbreviations needs to be standardised (eg: Met, TZD, BMD, BAP, u-NTX). Please do not capitalise generic drug names.
Answer: We have reviewed and revised the abbreviations to ensure that the same abbreviations are used throughout. In addition, we used professional language editing services to revise and embellish the full text. Thank you again for your support and suggestions on my manuscript.
Reviewer 2 Report (Previous Reviewer 3)
The author responded to questions and made revisions to the first version.
no more comment.
Author Response
Thank you again for your review and support of my manuscript.
Round 2
Reviewer 1 Report (Previous Reviewer 2)
Dear authors,
1. The revision is not satisfactory at this stage. The manuscript contains numerous typos which indicate a lack of proofreading efforts.
2. The entire Introduction should be restructured and divided into multiple paragraphs:
- the relationship between diabetes and bone health.
- controversies between diabetic medication and BMD.
- controversies between bone remodelling markers and BMD.
- justification for a meta-analysis.
- objective of the current review.
3. Do the authors consider circulating calcium as a bone remodelling marker? Its level is usually tightly regulated unless there is a disturbance of the parathyroid level.
4. why is there a 15 years time filter for literature search?
5. the search is up to May 2022, which is almost one year ago. The authors should update it.
6. Inclusion criteria: please reword (b) ... which include BMD or BTM as primary outcomes.
7. Is there a cut-off score for low-quality studies?
8. As I mentioned earlier, PRISMA checklist is not related to quality assessment. It should not be mentioned in that section.
9. Results: Exclusion results in the text should be followed by the number of excluded articles based on each criterion. E.g. review articles (n=7)
10. Again, despite using a random effect model, the authors still get large I2 values for most parameters. The authors should caution the readers about the results obtained.
Author Response
- The revision is not satisfactory at this stage. The manuscript contains numerous typos which indicate a lack of proofreading efforts.
Answer: We are very sorry that we did not do a good job of proofreading. We have re-proofread the manuscript and used a language retouching service to correct spelling and grammatical errors in the manuscript. The evidence of embellishment is provided in Annex I.
- The entire Introduction should be restructured and divided into multiple paragraphs:
- the relationship between diabetes and bone health.
- controversies between diabetic medication and BMD.
- controversies between bone remodelling markers and BMD.
- justification for a meta-analysis.
- objective of the current review.
Answer: Thank you very much for sorting out the content of the manuscript introduction more clearly. We have reorganized the structure of the introductory section and divided it into paragraphs as required to make the content clearer.
- Do the authors consider circulating calcium as a bone remodelling marker? Its level is usually tightly regulated unless there is a disturbance of the parathyroid level.
Answer: Thank you very much for your valuable comments, circulating calcium is susceptible to dietary and hormonal modulation, and the role of calcium in osteoporotic patients remains controversial. Therefore, it is not rigorous to use circulating calcium as a marker of bone remodeling. Therefore, we have removed this index from the manuscript.
- why is there a 15 years time filter for literature search?
Answer: On this issue, we initially considered setting a time limit of 15 years to include relatively new studies. However, given the comprehensive nature of the systematic review. We decided to remove the 15-year limit. We searched for all relevant randomized controlled trials to date. Regrettably, no additional studies were found.
- the search is up to May 2022, which is almost one year ago. The authors should update it.
Answer: The researchers re-searched the database, but no new relevant studies were published as of April 20, 2023.
- Inclusion criteria: please reword (b) ... which include BMD or BTM as primary outcomes.
Answer: Revised in the manuscript. (Page 6, Line 114-115)
- Is there a cut-off score for low-quality studies?
Answer: Low-quality studies are descriptive. For randomized trials, studies are usually considered credible if randomization was performed correctly, blinding was used, the allocation was concealed, objective endpoints were clear, and missing data were minimized. The opposite is considered low-quality research.
- As I mentioned earlier, PRISMA checklist is not related to quality assessment. It should not be mentioned in that section.
Answer: We apologize that we were not aware of this error and the PRISMA checklist has been removed from this section. (Page 7, Line 137-140)
- Results: Exclusion results in the text should be followed by the number of excluded articles based on each criterion. E.g. review articles (n=7)
Answer: We indicated the specific number after the exclusion of literature. (Page9, Line 167-172)
- Again, despite using a random effect model, the authors still get large I2 values for most parameters. The authors should caution the readers about the results obtained.
Answer: We have added the following to the limitations section of the manuscript " In the heterogeneity test, BMD, CTX, and P1NP showed high heterogeneity (I2>50%). Even though we used a random effects model to reduce the effect of heterogeneity, the effect of higher heterogeneity on the results of the meta-analysis could not be eliminated due to the small number of included studies. Therefore, we should interpret these observations with caution and look forward to more high-quality clinical studies in the future to remedy this deficiency." to remind the reader to refer to these conclusions with caution. (Page 18, Line 336-341)

Round 3
Reviewer 1 Report (Previous Reviewer 2)
I have no further comments. The authors have addressed my concerns.
This manuscript is a resubmission of an earlier submission. The following is a list of the peer review reports and author responses from that submission.
Round 1
Reviewer 1 Report
Comments:
- - Introduction, last paragraph: “We also analyzed blood calcium, bone-specific alkaline phosphatase (BAP) and urine n-telopeptide of type I collagen (u-NTX), which were associated with a reduced risk of bone mineral density.”
What did it mean? I think the author means: “We also analyzed blood calcium, bone-specific alkaline phosphatase (BAP) and urine n-telopeptide of type I collagen (u-NTX), which were negatively correlated with bone mineral density.”
- - Introduction, last paragraph: Bone turnover markers are helpful in assessing response to treatment in osteoporosis.
Reviewer 2 Report
The authors conducted a meta-analysis comparing the skeletal effects of metformin and thiazolidinediones in diabetic patients. They summarized the findings of 7 studies and found that metformin increased BMD compared to thiazolidinediones up to 52 weeks but decreased BMD for longer period. Metformin also decreased PINP and CTX levels compared to thiazolidinediones. The authors concluded that metformin inhibited both osteogenesis and osteolysis, and TZDs 27 inhibited osteogenesis but promoted osteolysis.
As high bone remodelling (increased bone formation and increased bone resorption) can be a characteristic of osteoporosis, the suppression of PINP and CTX by metformin could reflect an inhibition of the high bone remodelling phenomenon. It is difficult to conclude, based on the existing observations, that metformin prevent osteogenesis and osteolysis. At best, we can say that metformin is not bone anabolic.
The interesting observation of metformin increased BMD and later decreased BMD is not being discussed sufficiently. I hope the authors can elaborate on that.
In addition, the discrepancies between the results of PINP/CTX and BAP/urine NTX should be discussed.
Minor comments:
Line 205: metformin can work as a receptor activator of nuclear factor kappa-B ligand “inhibitor?” – I suspect a missing word.
Line 225: what is “endodontic animals”? – do you mean “animals with endodontic diseases”?
Reviewer 3 Report
This review comparison of the effects of metformin (MET) and thiazolidinediones (TZDs) on bone metabolism via meta-analysis. Seven studies involving 1,656 patients were eventually included. After reviewing this data, the authors' conclusions were that MET inhibited, and TZDs inhibited osteogenesis but promoted osteolysis. The manuscript writing is well, and easy to read. However, the conclusion still needs to discuss.
Comments:
1. I do not think the authors got 1656 patients' original raw data. All conclusion in this manuscript is from seven published original articles mentioned in this manuscript, so to use meta-analysis I think is not correct.
2. The study design of the original articles did not descript clearly in this manuscript.
In Bilezikian (published in 2013, not in 2012) article, firstly, the study design is a 52-week double-blind treatment with RSG or MET, after that, a 24-week open-label follow-up phase in which RSG was discontinued and all subjects received MET.
The data from Mori (2017) and Kanazawa (2010) are from 3 months study; Zinman (2010) data is from 12 months study; Lierop data from 24 weeks. All this information should be clearly described.
3. The results from the original articles did not correctly describe in this manuscript.
a. In Bilezikian article, from baseline to week 52, femoral neck, total hip, and lumbar spine mean BMD was decreased in RSG treated group. After switching to MET treatment 24 weeks for previously RSG patients, at week 76, the decreased BMD was an attenuated decrease, even a slight increase trend based on decreased BMD levels.
MET-treated group, the femoral neck, and lumbar spine BMD were slightly increased, except total hip BMD was decreased. At week 76, the BMD was little change in femoral neck BMD, or increased in lumbar spine BMD, and a minimal continuous decrease in total hip BMD.
b. Miller (2016) data is hip cortical vBMD (volumetric bone mineral density, measured by peripheral quantitative computed tomography. Bilezikian (2013) is real BMD, measured by DXA. This information should be clearly described in the manuscript.
4. In Figure 3 of Bilezikian original article, one incorrectly written for MET-treated patients in the open-label MET phase, (+1.03 incorrectly written as -1.03), the same incorrectly was copied in this manuscript in figure 4.
5. In my opinion, the figures (figures 4, 5, and 6) in this manuscript are more complex, it will be better to directly display the original article's data.
6. Bilezikian original article also has CTX, PINP, and BAP data, which was not mentioned in this manuscript, why?